# Association of Computer Vision Syndrome with Depression/Anxiety among Lebanese Young Adults: The Mediating Effect of Stress

**DOI:** 10.3390/healthcare11192674

**Published:** 2023-10-02

**Authors:** Rita Issa, Michel Sfeir, Vanessa Azzi, Pascale Salameh, Maria Akiki, Marwan Akel, Souheil Hallit, Sahar Obeid, Diana Malaeb, Rabih Hallit

**Affiliations:** 1School of Medicine and Medical Sciences, Holy Spirit University of Kaslik, Jounieh P.O. Box 446, Lebanon; rita.j.issa@net.usek.edu.lb (R.I.); vanessakazzi18@outlook.com (V.A.); souheilhallit@hotmail.com (S.H.); hallitrabih@hotmail.com (R.H.); 2Faculty of Social and Political Sciences (SSP), Institute of Psychology (IP), University of Lausanne, 1015 Lausanne, Switzerland; michelsfeir@protonmail.com; 3School of Medicine, Lebanese American University, Byblos 5053, Lebanon; pascalesalameh1@hotmail.com; 4INSPECT-LB (Institut National de Santé Publique, d’Épidémiologie Clinique et de Toxicologie-Liban), Beirut P.O. Box 12109, Lebanon; marwan.akel@liu.edu.lb; 5Medical School, University of Nicosia, 2417 Nicosia, Cyprus; 6Faculty of Pharmacy, Lebanese University, Hadat 1103, Lebanon; 7Department of Internal Medicine, Saint Michael’s Medical Center, Newark, NJ 07102, USA; maria.s.akiki@net.usek.edu.lb; 8School of Pharmacy, Lebanese International University, Beirut P.O. Box 146404, Lebanon; 9Applied Science Research Center, Applied Science Private University, Amman 11931, Jordan; 10Research Department, Psychiatric Hospital of the Cross, Jal Eddib P.O. Box 60096, Lebanon; 11Social and Education Sciences Department, School of Arts and Sciences, Lebanese American University, Jbeil P.O. Box 13-5053, Lebanon; saharobeid23@hotmail.com; 12College of Pharmacy, Gulf Medical University, Ajman P.O. Box 4184, United Arab Emirates; 13Department of Infectious Disease, Notre-Dame des Secours University Hospital, Byblos Postal Code 3, Lebanon; 14Department of Infectious Disease, Bellevue Medical Center, Mansourieh P.O. Box 295, Lebanon

**Keywords:** computer vision syndrome, depression, anxiety, stress, adults, Lebanon

## Abstract

Computers have become indispensable in daily activities. With this excess use of electronics, computer vision syndrome (CVS), a highly prevalent condition, is associated with various symptoms. Although understanding the relationship between CVS and mental health has been reported, the impact of CVS has not been explored on more than one psychological aspect. We hypothesize that higher CVS symptoms could be associated with higher levels of anxiety and depression, mediated by higher stress. Therefore, the objective of this study was to determine the association between CVS and depression and anxiety among a sample of Lebanese young adults, along with evaluating the mediating effect of stress on these associations. Between August 2020 and April 2021, 749 participants completed an online questionnaire for this cross-sectional study. Females compared to males (Beta = 3.73) and those with CVS compared to those who did not (Beta = 3.14) were significantly associated with more anxiety, whereas having a university level of education compared to secondary or less (Beta = −3.02) was significantly associated with less anxiety. Females compared to males (Beta = 2.55) and those with CVS compared to those without (Beta = 2.61) were significantly associated with more depression, whereas being of an older age (Beta = −0.18) was significantly associated with less depression. Stress partially mediated the association between CVS and anxiety and between CVS and depression. More CVS was significantly associated with more stress (Beta = 3.05). Higher stress was significantly associated with more anxiety (Beta = 0.70) and depression (Beta = 0.71), whereas more CVS was significantly and directly associated with more anxiety (Beta = 3.14) and depression (Beta = 2.61). This study is the first worldwide to evaluate an association between CVS and mental health. Our results serve as a starting point for healthcare providers (psychiatrists and psychologists, most importantly) to look deeper into CVS when looking for reasons behind mental health issues. Further studies are warranted to confirm our results and look for more factors and mediators in such associations.

## 1. Introduction

Depression and anxiety are the most common psychiatric illnesses, with more than 300 million people in the world affected by depression and almost 7.3% of the world’s population, or one out of nine people having at least one anxiety disorder [1,2,3,4]. In fact, people with one disorder can also have the features of a second one as a comorbidity [5]; however, distinguishing them is essential and has been proven to be difficult [6]. Depression, a mood disorder, is a state of lacking interest in daily activities with an incapacity to appreciate life [7]. It affects mental and physical health and constitutes the primary cause of disability in the world [8], thus influencing quality of life [9,10,11]. On another hand, anxiety is characterized by feelings of danger and tension; it consists of emotional, cognitive, and physiological responses linked to a future event that may consist of a threat [12]. Nevertheless, despite the tremendous improvement in depression and anxiety treatment, their prevalence remains high [13]. The pandemic was associated with more mental health issues [14]; in fact, symptoms of depression and anxiety increased during the pandemic worldwide by 27.6% and 25.6%, respectively [15].

Several factors have been shown to be correlated with depression and anxiety; age and sex are linked to these disorders, with a female/male ratio of 1.73:1 for anxiety disorder [16] and 1.71:1 for major depression [17] and a high incidence among the elderly [18]. Higher education levels can have a double effect, where, in some cases, it protects against depression, but in other cases, such as medical students, depressive symptoms are more likely to be found [19,20]. Socioeconomic status (SES) has a great impact on stress; people with low SES seem to be more exposed to stressors and mental health problems than those with high SES [21].

In addition, stress has been marked as one of the important factors affecting mental health [22,23]. Any type of stress exposure can be associated with a decrease in both physical and mental health [24].

Computers have become indispensable in daily activities and an integral part of everyday tasks. Nowadays, electronic devices, including computers, are utilized in offices, schools, and universities as their main companion. With this excess use of electronics, ocular, visual, and musculoskeletal symptoms have been reported, including eyestrain, dryness, irritation, burning, blurring, double vision, ocular strain and ache, neck and shoulder pain, redness, etc. These symptoms represent the computer vision syndrome (CVS) [25]. During the declaration of the coronavirus as a global disease, governments took measures to restrain the outbreak. Lockdowns were exerted, and online activities/learning platforms were initiated to ensure optimum interactions between groups. The lockdown has led to an increase in screen time, influencing the wellbeing of users [26,27]. The excessive use of electronic devices represents a major health concern, given the fact that approximately 80% of people’s daily activities involve the use of such devices [28]. Excessive computer use for extended hours is linked to depression and anxiety, as well as headaches [29] and a reluctance to wake up the following morning [30]. Data from the previous literature have shown that the use of devices can lead to psychological/psychiatric problems, such as depression or anxiety, due to the pain and dryness caused by eye damage [31,32]. In addition, CVS deteriorates sleep quality, which, in turn, triggers psychological disorders [32]. 

CVS, a recent and highly prevalent condition, was recently assessed for associated risk factors, long-term effects, and related problems [33,34,35]. However, our study is unique and the first of its kind worldwide to evaluate an association between CVS and mental health, with stress being a mediator in these associations. Analyzing the effect of stress is extremely important since if addiction to electronic devices develops, stress proves a powerful mediator [36]. We hypothesize that higher CVS symptoms are associated with higher levels of anxiety and depression mediated by higher stress. In the absence of similar studies, we recommend further studies to assess these associations in more depth. 

Furthermore, most studies were conducted in the Western region and examined global prevalence, but there has been a lack of studies conducted in the Middle East that assessed the influence of CVS on mental health [37,38]. Therefore, the objective of this study is to determine the association between CVS and depression and anxiety among a sample of Lebanese young adults, along with evaluating the mediating effect of stress in these associations.

## 2. Methods

### 2.1. General Study Design

This cross-sectional study, enrolling 749 participants, was conducted from August 2020 to April 2021, addressing digital device users in all of Lebanon’s districts (Mount Lebanon, Beirut, North, South, and Bekaa) with an online questionnaire. The snowball sampling technique was employed to select participants for our research. To facilitate data collection, we developed a digital version of the questionnaire using the Google Forms software, opting for an online approach for ease and accessibility. Before participation, all participants were informed about this study’s objectives and provided with clear instructions for completing the questionnaire through an online platform. We rigorously examined internet protocol (IP) addresses to ensure data integrity to prevent duplicate survey submissions and maintain data validity. Notably, no incentives or rewards were offered to participants, ensuring that responses were entirely voluntary. Thus, participants were asked to send this link to others using a digital device. The questionnaires were written in Arabic, and the link was sent via WhatsApp and email to participants.

### 2.2. Sample Size Calculation

A sample size of 395 was required, and these results were obtained using G-Power software 3.0.10 based on an effective size f2 = 2%, an alpha error of 5%, a power of 80%, and considering 10 factors included in the multivariate analysis.

### 2.3. Questionnaire

Firstly, and prior to beginning data collecting, pilot research involving 10 participants was conducted. Unclear questions were explained, and a few linguistic changes were made. The final database does not contain the information collected from those 10 individuals. Informed consent was included in the first section of the questionnaire, which was composed of closed and semi-open questions. Participants were assured of the anonymity of their responses and informed about the importance of offering their informed consent before participation. The remaining questions were accessible to the participants after their approval. In the second section, the sociodemographic details of the participants, such as their age, gender, region, level of education, occupation, and home crowding index, were evaluated. The home crowding index was computed by dividing the number of occupants in the home by the number of rooms (excluding bathrooms and kitchens), thus reflecting the socioeconomic status of participants [38].

The total number of hours spent using a computer was estimated by multiplying the total number of years by the number of days of computer use per week and by the total number of hours used per day.

In the fourth part of the questionnaire, we added the following scales:

#### 2.3.1. Computer Vision Syndrome Scale

The Computer Vision Syndrome Scale, developed by Segui et al. [39], was used to determine the frequency and severity of 16 symptoms associated with inappropriate computer use. Each symptom was treated as a separate item in the questionnaire. According to Cronbach’s alpha (0.78), the participant with a total score greater than or equal to six is said to have computer vision syndrome. One psychologist translated the CVS scale into Arabic, while a second psychologist translated it back into English. A consensus was reached to reconcile the differences between the two English translations.

#### 2.3.2. Beirut Distress Scale (BDS-10)

Utilizing BDS-10, stress was evaluated. It was used to evaluate how distressed computer users were psychologically. It is a 10-question scale with ratings on a 5-point Likert scale, ranging from 0 (never) to 4 (always), with higher ratings indicating greater stress [40] (Cronbach’s alpha = 0.821).

#### 2.3.3. Lebanese Anxiety Scale

The Lebanese Anxiety Scale (LAS) has ten items, seven of which range from 0 (no symptoms present) to 4 (extremely severe symptoms), and three of which range from 0 to 3. Higher scores indicate greater anxiety. This scale has been validated using adults and teenagers [41,42]. Cronbach’s alpha in this study was 0.939.

#### 2.3.4. Montgomery–Asberg Depression Rating Scale (MADRS)

Validated in Lebanon [39], this scale is composed of 10 items scored from 0 to 6, with higher scores indicating more severe depressive symptoms [43]. In this study, Cronbach’s alpha was 0.831.

### 2.4. Ethical Consideration

The ethical approval for this research was given by the Ethics and Research Committee of the Psychiatric Hospital of the Cross (HPC-034-2020). All participants provided their informed consent by submitting an online form. All research steps were conducted in accordance with the relevant guidelines and regulations.

### 2.5. Statistical Analysis

Version 23 of the statistical analysis program SPSS was used. According to gender and educational levels, the general population was weighted. Given that the values for skewness and kurtosis ranged from −1 to +1, the LAS and MADRS scores showed a normal distribution [44]. The assumptions of normality in samples bigger than 300 were strengthened by these circumstances [45]. As a result, Student’s *t*-test was employed to determine whether there was a correlation between the LAS and MADRS scores and dichotomous factors (i.e., gender and married status), whereas the ANOVA test was employed to compare three or more means (i.e., educational attainment and monthly income). Two continuous variables (age, BMI, depression, anxiety, and stress) were correlated using the Pearson correlation test. After adjusting for potential confounding factors, such as age, gender, the house crowding index, education level, stress, and the cumulative number of hours spent on a computer, a multivariate analysis of covariance (MANCOVA) was performed using the anxiety and depression scores as dependent variables and the presence or absence of CVS as the primary independent variable.

Three paths were calculated using the PROCESS SPSS Macro version 3.4 model four [46]. Pathway A calculated the regression coefficient for the relationship between the presence of CVS and stress, Pathway B looked at the relationship between stress and LAS/MADRS scores without considering the effect of CVS, and Pathway C estimated the overall and direct impact of the presence of CVS on LAS/MADRS scores. The effects of the indirect intervention were computed using pathway AB. The bias-corrected bootstrapped 95% confidence intervals created by the macro should not pass by zero when testing the significance of the indirect impact [46]. To create parsimonious models, all covariates with effect sizes or correlations greater than │0.24│ in the bivariate analysis were incorporated as independent variables in the multivariable and mediation models [17]. A *p* < 0.05 was considered significant.

## 3. Results

### 3.1. Sociodemographic and Other Characteristics of the Participants 

A total of 749 participants were enrolled in this study. The mean age of the participants was 24.51 ± 7.68 years, with 65.6% females. In addition, 528 (70.5%) of the participants had computer vision syndrome. Other details can be found in Table 1.

### 3.2. Bivariate Analysis

Significantly higher depression and anxiety were found in females compared to males, in those with a secondary level of education compared to the other levels, and in those with CVS compared to those who did not (Table 2). Furthermore, higher stress was significantly associated with higher anxiety and depression scores, whereas a higher cumulative number of hours of computer work was significantly associated with less anxiety and depression. Older age was significantly associated with less depression, whereas a higher household crowding index was significantly associated with less anxiety and more depression (Table 3).

### 3.3. Adjusted Means

Higher mean anxiety (14.98 vs. 12.08) and depression (14.2 vs. 11.74) scores were significantly found in participants with CVS compared to those without CVS (Figure 1). Results were adjusted over the potentially confounding variables: age, gender, house crowding index, education level, and the cumulative number of hours of computer work.

### 3.4. Multivariate Analysis (MANCOVA)

Females compared to males (Beta = 3.73) and those with CVS compared to those without (Beta = 3.14) were significantly associated with more anxiety, whereas having a university level of education compared to secondary or less (Beta = −3.02) was significantly associated with less anxiety (Table 4, Model 1).

Females compared to males (Beta = 2.55) and those with CVS compared to those without (Beta = 2.61) were significantly associated with more depression, whereas being of older age (Beta = −0.18) was significantly associated with less depression (Table 4, Model 2).

### 3.5. Mediation Analysis

A mediation analysis was conducted, taking the presence/absence of CVS as the independent variable, stress as a mediator, and anxiety/depression as the dependent variables. The results show that stress mediated the association between CVS and anxiety and between CVS and depression (Table 5, Figure 2 and Figure 3).

## 4. Discussion

This is the first study conducted among the Lebanese population to evaluate the association between computer vision syndrome and mental health. Our pursuit of this topic is backed up by the fact that recent studies have pointed to a significant rise in CVS around the world during the past few years, from 64% to 90% [37,38], thus sparking our interest in evaluating the potential association between CVS and mental health. This current study was conducted to determine the association between CVS and depression and anxiety among a sample of Lebanese young adults, along with evaluating the mediating effect of stress in these associations. After computing all the data collected, our results clearly state that the presence of CVS is significantly associated with more anxiety and depression.


*Computer vision syndrome and depression*


Our results show that CVS is associated with depression, which is consistent with the results from another study that showed a positive association between the mean level of computer use and depression [40]. In fact, computer use was significantly associated with depression and with a higher likelihood of experiencing depressive symptoms if the time spent in front of the screen exceeded 6 h per day [41]. Screen time was also positively correlated with anxiety and depressive symptoms, as well as insomnia [27,42,43]. An increase in computer usage has been correlated with more depression when someone spends more than 30 h per week on their device [44]. Our results concur with those of another study that connected excessive use of technology to depression among students [32]. Our results can be explained by the fact that depression may be linked to the subjective symptoms of dry eye illness rather than objective symptoms [45]. According to other studies, there are various interconnected factors that mediate the association between CVS and depression, such as sleep disruption and reduced physical activity. 

Our study results suggest that women experience more depression and anxiety compared to males, which aligns with the findings of previous studies [41,46]. Such gender differences in the prevalence of depression can be explained by the fact that women experience hormonal fluctuations regularly and particularly during puberty, prior to menstruation, following pregnancy, and at perimenopause, which may be a trigger for depression [17]. Specifically, there is a biological reason for higher depression among females correlating with higher levels of inflammatory, neurotrophic, and serotonergic markers and the direct relation between some inflammatory and neurotrophic markers and the severity of depressive symptoms [47]. Additionally, females are more vulnerable to environmental factors that predict the development of depression, such as discrepancies in employment, income rates, education, and psychological stress [48,49].


*Computer vision syndrome and anxiety*


Our results show that CVS is associated with anxiety, which can be explained by the fact that the dry eye symptom in computer vision syndrome may be associated with symptoms of anxiety, which is related to work or even depression [50]. Additionally, we hypothesize that excessive computer/screen use might be associated with more physical symptoms (i.e., ocular problems, low back pain, headaches, etc.), which, in turn, might be related to more mental health issues [30]. We are aware that computer vision syndrome is different from excessive computer use and screen time use; however, to our knowledge, our study is the first to show an association between CVS, depression, and anxiety, with a focus on the effect of stress as a mediator. A prior study found that the presence of CVS was connected to higher anxiety levels, which can be explained by the connection between anxiety and physical discomfort due to eye illnesses, headaches, and neck and shoulder pain [45]. Further studies have linked the association between CVS and anxiety due to cognitive overload [51]. The extended use of computers requires extensive concentration and tremendous attention, which can contribute to mental fatigue and trigger feelings of anxiety. In addition, the heavy use of electronic devices disrupts sleep patterns, increases irritability, and heightens anxiety [29]. Also, anxiety can be attributed to the lack of physical activity and relaxation caused by prolonged digital device use [52]. Previous studies have shown that females have a higher severity of anxiety, as well as other anxiety disorders such as social phobia [53,54]. On a neurobiological level, females revealed a high level of hypothalamic-pituitary-adrenal (HPA) in stressful situations, as well as in non-stressful ones [55]. In addition, females showed more corticosterone responses in anxiety tests (for example, the elevated plus-maze) [56]. Furthermore, it has been highlighted that females believe that worry is uncontrollable and must be avoided due to metacognition [57]. Thus, females think that worry is uncontrollable, which affects their health and leads to anxiety. The findings of this study show that having a university level of education compared to secondary or less is correlated with less anxiety, which aligns with a previous study that supported the idea that higher education can play a protective factor against anxiety [58].


*Computer vision syndrome and stress*


Our investigation demonstrates that the relationships between CVS and depression/anxiety are mediated by stress. The effects of stress on the body over time have been linked to psychiatric diseases such as depression, anxiety, and brief sleeplessness, according to research [55]. On the other hand, adolescents who are under stress may become irritable and angry more frequently [56], and adults who have had unfavorable experiences may become more aggressive [32]. It has been reported that stress mediates the association between CVS, depression, and anxiety through physiological, psychological, and behavioral responses [59]. Stress is considered a well-known trigger for the release of stress hormones such as cortisol, which disrupts the normal physiological hypothalamus–pituitary axis function, including sleep regulation [60]. These abnormalities contribute to physical discomfort, which is associated with CVS, and unstable moods. Furthermore, stress exerts an effect on the cellular level manifested by the release of vasoconstrictors that affect blood pressure, heart rate, and glucose levels, which all exert an effect on cognitive function, potentially deteriorating depression and anxiety [61]. In Lebanon, individuals who followed news about COVID-19 and those in contact with patients who have COVID-19 showed a high level of stress [14]. In addition, the economic crisis hitting Lebanon since October 2019 has led to high unemployment rates reaching 40% [62]. A recent Lebanese study showed that higher monthly incomes were shown to be associated with less stress [14]. During the pandemic, people were more prone to experiencing mental health disorders, like stress, depression, or anxiety, since they were using the internet, social media, computers, and smartphones more frequently because of the lockdowns imposed by governments worldwide during the pandemic, the shift to online work in corporations and academia, and to avoid feelings of loneliness [48,63].

### 4.1. Clinical Implications

Our results show that CVS is associated with psychiatric disorders, such as depression, anxiety, and insomnia. These significant associations between CVS and mental health may have important clinical implications for adult’s well-being and development. These results indicate the importance of early screening and assessments to identify individuals at risk for both physical and mental disorders. Thus, early intervention can help halt the development of mental disorders and provide necessary care. Furthermore, raising awareness about the potential detrimental effects of digital devices is vital for implementing proactive strategies. In addition, collaboration is needed at all levels, especially between healthcare professionals and patients, to address the health status and provide holistic support.

### 4.2. Limitations

Some limitations of this study should be acknowledged. A residual confounding bias is possible since many factors known to affect depression and anxiety were not cited in our study, such as genetic predisposition [49], unhealthy lifestyles (smoking, drinking) [64,65], insomnia [66], being obese or underweight [67], and some chronic medical diseases, such as cardiac illnesses, diabetes, or cancer [67], and also types of computer devices. This survey was conducted during the COVID-19 pandemic when responders might accordingly be prone to more depression and anxiety [14], and thereby, their scores could be biased (information bias since participants might over or under-report their symptoms). Moreover, those symptoms were not diagnosed by a clinician. A selection bias is present because of the way data were collected (snowball technique); therefore, data are not generalizable to the whole population. Furthermore, our study population was homogenous (comprising solely Lebanese individuals) and used the snowball sampling technique, therefore limiting the generalizability of these findings to larger populations or different demographic groups. Finally, the Computer Vision Syndrome Scale used in this study has not been validated in Lebanon yet.

## 5. Conclusions

Despite all these limitations, this study is the first worldwide to evaluate an association between CVS and mental health, with stress being a mediator for those associations. Our results serve as a starting point for healthcare providers (psychiatrists and psychologists, most importantly) to look deeper into CVS when looking for reasons behind mental health issues. Further studies are warranted to confirm our results and assess other factors, such as the total time spent using any kind of digital screen, the type of electronic devices, and the reasons for utilizing the device, which might be considered a vital factor that mediates the association between CVS and mental health.

## Figures and Tables

**Figure 1 healthcare-11-02674-f001:**
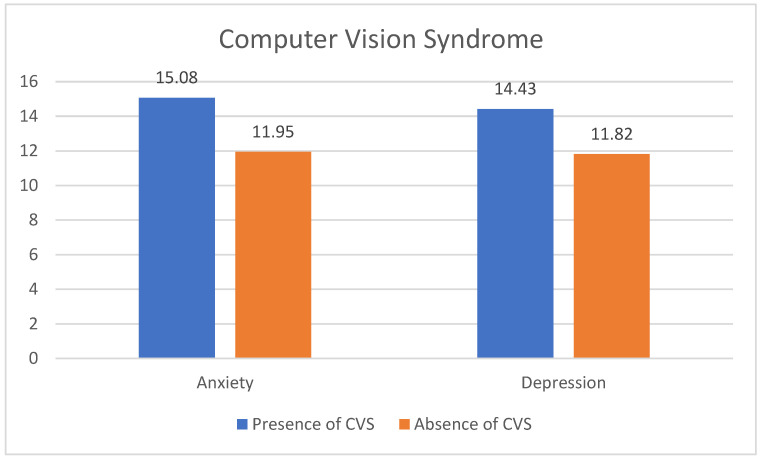
Comparison of the mean migraine and insomnia scores depending on the presence/absence of computer vision syndrome after adjustments for sociodemographic characteristics. Anxiety: *p* = 0.001; Depression *p* = 0.013, CVS: computer vision syndrome.

**Figure 2 healthcare-11-02674-f002:**
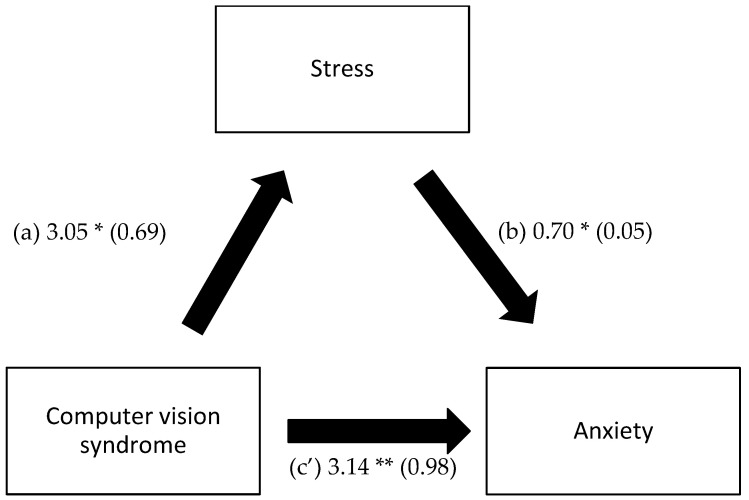
(**a**) Relation between computer vision syndrome and stress; (**b**) Relation between stress and anxiety; (**c’**) Relation between computer vision syndrome and anxiety. Numbers are displayed as regression coefficients (standard error). * *p* < 0.001; ** *p* < 0.01.

**Figure 3 healthcare-11-02674-f003:**
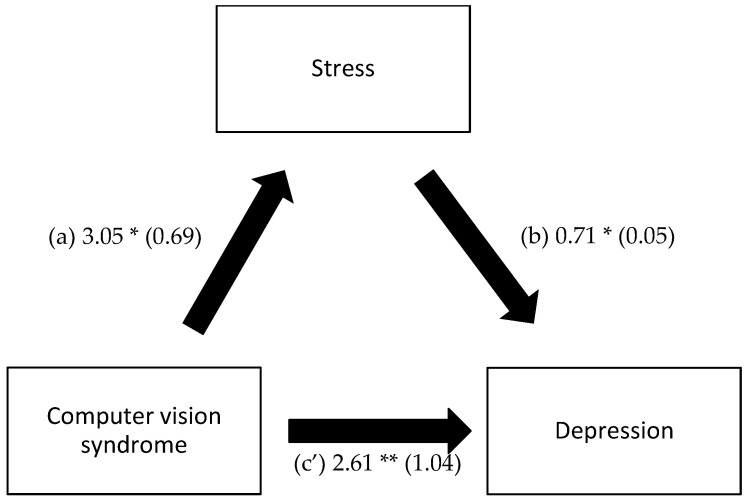
(**a**) Relation between computer vision syndrome and stress; (**b**) Relation between stress and depression; (**c’**) Relation between computer vision syndrome and depression. Numbers are displayed as regression coefficients (standard error). * *p* < 0.001; ** *p* < 0.05.

**Table 1 healthcare-11-02674-t001:** Sociodemographic and other characteristics of the participants.

Variable	N (%)
Gender	
Male	258 (34.4%)
Female	491 (65.6%)
Education level	
Complementary or less	21 (2.8%)
Secondary	103 (13.8%)
University	625 (83.4%)
	Mean ± SD
Age (in years)	24.51 ± 7.68
Depression	13.12 ± 10.23
Anxiety	13.48 ± 9.66
Stress	11.82 ± 7.09
Household crowding index	1.05 ± 0.48

**Table 2 healthcare-11-02674-t002:** Bivariate analysis of categorical factors associated with depression and anxiety.

Variable	Anxiety	*p*	Effect Size	Depression	*p*	Effect Size
Gender		**<0.001**	0.551		**<0.001**	0.312
Male	11.48 ± 10.22			11.39 ± 12.33		
Female	16.89 ± 9.41			15.00 ± 10.78		
Education level		**0.011**	0.241		0.315	0.088
Secondary or less	15.49 ± 10.09			13.95 ± 11.80		
University	13.12 ± 9.56			12.99 ± 9.90		
Computer vision syndrome		**0.027**	0.235		**0.001**	0.338
No	14.51 ± 10.42			12.09 ± 9.78		
Yes	16.82 ± 9.16			15.66 ± 11.29		

Numbers in bold indicate significant *p*-values.

**Table 3 healthcare-11-02674-t003:** Bivariate analysis of continuous variables associated with depression and anxiety.

Variable	Anxiety	Depression
Stress	r = 0.443; ***p* < 0.001**	r = 0.543; ***p* < 0.001**
Age	r = 0.067; *p* = 0.068	r = −0.114; ***p* = 0.002**
Household crowding index	r = −0.092; ***p* = 0.012**	r = 0.086; ***p* = 0.019**
Cumulative number of hours of computer work	r = −0.121; ***p* = 0.001**	r = −0.129; ***p* < 0.001**

Numbers in bold indicate significant *p*-values.

**Table 4 healthcare-11-02674-t004:** Multivariable analysis of covariance (MANCOVA).

Model 1: LAS Score as the Dependent Variable.				
Variable	Beta	*p*	95% CI	Partial Eta Squared
Age	−0.05	0.387	−0.15–0.06	0.001
Household crowding index	0.32	0.678	−1.19–1.83	0.001
Cumulative number of hours of computer work	0.001	0.882	−0.002–0.002	0.001
Gender (females vs. males *)	3.73	**<0.001**	2.17–5.28	0.034
Education level (university vs. secondary or less *)	−3.02	**0.004**	−5.08–−0.96	0.013
CVS (yes vs. no *)	3.14	**0.001**	1.22–5.05	0.016
**Model 2: MADRS Score as the Dependent Variable.**				
**Variable**	**Beta**	** *p* **	**95% CI**	**Partial Eta Squared**
Age	−0.18	**0.002**	−0.29–−0.07	0.015
Household crowding index	0.26	0.755	−1.36–1.87	0.001
Cumulative number of hours of computer work	0.001	0.637	−0.002–0.003	0.001
Gender (females vs. males *)	2.55	**0.003**	0.89–4.21	0.014
Education level (university vs. secondary or less *)	−1.57	0.161	−3.78–0.63	0.003
CVS (yes vs. no *)	2.61	**0.013**	0.56–4.65	0.010

* Reference category; Numbers in bold indicate significant *p*-values; CI = confidence interval; CVS = computer vision syndrome. LAS: Lebanese Anxiety Scale, MADRS: A new depression scale–Asberg Depression Rating Scale.

**Table 5 healthcare-11-02674-t005:** Mediation analysis.

	Direct Effect			Indirect Effect		
	Beta	SE (B)	BCa CI	Beta	SE (B)	BCa CI
Anxiety	0.99	0.86	−0.69–2.68	2.14	0.50	1.18–3.16 *
Depression	0.44	0.93	−1.39–2.27	2.16	0.51	1.19–3.18 *

* indicates significant mediation. SE (B): standard error for unstandardized beta.

## Data Availability

All data generated or analyzed during this study are included in this published article.

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
