# Peer review of "Association of Computer Vision Syndrome with Depression/Anxiety among Lebanese Young Adults: The Mediating Effect of Stress"

_healthcare, 2023, doi:10.3390/healthcare11192674_

Round 1
Reviewer 1 Report
In the current study, the authors set out to investigate the relationship between Computer Vision Syndrome (CVS) and depression and anxiety within a cohort of Lebanese young adults, while concurrently examining the mediating role of stress in these associations. The chosen research focus is undeniably captivating and warrants commendation for its rigorous execution. The writing in the study is precise and apt, facilitating a lucid comprehension of the research. Please, find enclosed some commentaries.
- Numerical identifiers should be added to the section titles to enhance clarity and organisation.
- In the "Methods" section, specifying whether the questionnaires were administered anonymously or if the investigators were masked to the responses would be valuable. Additionally, it is advisable to clarify whether information regarding the types of computer devices used was included in the questionnaire, or if this aspect was addressed as a generic point.
- In the "Limitations" section, it is crucial to acknowledge that the questionnaires used in the study had not been previously validated in Arabic by other research endeavours. Furthermore, it should be mentioned that while the study's population was homogenous (comprising solely Lebanese individuals), the findings are limited to a specific geographic region.
- Regarding tables and figures, it is imperative that acronyms employed therein be consistently reported in the corresponding table/figure captions for clarity and reference.
Author Response
In the current study, the authors set out to investigate the relationship between Computer Vision Syndrome (CVS) and depression and anxiety within a cohort of Lebanese young adults, while concurrently examining the mediating role of stress in these associations. The chosen research focus is undeniably captivating and warrants commendation for its rigorous execution. The writing in the study is precise and apt, facilitating a lucid comprehension of the research. Please, find enclosed some commentaries.
- Numerical identifiers should be added to the section titles to enhance clarity and organisation.
All numerical identifiers are added to each section.
- In the "Methods" section, specifying whether the questionnaires were administered anonymously or if the investigators were masked to the responses would be valuable. Additionally, it is advisable to clarify whether information regarding the types of computer devices used was included in the questionnaire, or if this aspect was addressed as a generic point.
We added the following to the questionnaire part: Participants were assured of the anonymity of their responses and informed about the importance of offering their informed consent before participation.
We added the idea as a limitation: Residual confounding bias is possible since many factors known to affect depression and anxiety were not cited in our study, such as genetic predisposition [54], unhealthy lifestyles (smoking, drinking) [55, 56], insomnia [57], being obese or underweight [58], and some chronic medical diseases such as cardiac illnesses, diabetes, cancer [58], also types of computer devices.
- In the "Limitations" section, it is crucial to acknowledge that the questionnaires used in the study had not been previously validated in Arabic by other research endeavours. Furthermore, it should be mentioned that while the study's population was homogenous (comprising solely Lebanese individuals), the findings are limited to a specific geographic region.
We added the following limitations: Furthermore, our study population was homogenous (comprising solely Lebanese individuals) and using the snowball sampling technique, therefore limiting the generalizability of the findings to larger populations or different demographic groups. Finally, the Computer Vision Syndrome Scale used in this study has not been validated in Lebanon yet.
- Regarding tables and figures, it is imperative that acronyms employed therein be consistently reported in the corresponding table/figure captions for clarity and reference.
All the captions are added below each table:
Table 4: LAS: Lebanese Anxiety Scale, MADRS: Montgomery-Asberg Depression Rating Scale.
Table 5: SE(B): standard error for unstandardized beta.
Reviewer 2 Report
Reviewer's Report for Manuscript: "Association of Computer Vision Syndrome with Depression/Anxiety among Lebanese Young Adults: The Mediating Effect of Stress"
Abstract:
The abstract provides a comprehensive overview of the study's objectives, methodology, and findings. However, there are a few areas that need improvement:
1. Lack of Clarity in Research Gap: While the abstract highlights the significance of the study in exploring the association between Computer Vision Syndrome (CVS) and mental health issues, the abstract fails to articulate the research gap clearly. Given the existing literature and what specific knowledge gap it addresses, it should explicitly state why this study is important.
2. Absence of Hypotheses: The abstract does not mention any hypotheses or expected outcomes. Including these would give readers a better understanding of the study's direction and focus.
3. Incomplete Mention of Findings: While the abstract discusses the associations between CVS, depression, anxiety, and stress, it lacks specific information on the strength of these associations and the extent of the mediating effect of stress. Quantitative results or effect sizes would enhance the abstract's clarity and impact.
Introduction:
The introduction provides background information about Computer Vision Syndrome (CVS) and its potential impact on mental health. However, some issues need to be addressed:
1. Limited Literature Review: The introduction lacks a comprehensive review of existing literature on CVS, mental health, and stress. A more detailed overview of relevant studies would provide a stronger foundation for the study's significance and research questions.
2. Research Objectives and Hypotheses: While the study's objectives are briefly mentioned, they could be more clearly stated. Additionally, hypotheses should be explicitly outlined to guide readers in understanding the study's purpose and direction.
3. Link to the Research Gap: The introduction should emphasize the unique contribution of this study in addressing the research gap. How does this study advance our understanding beyond previous research? This connection needs to be explicitly established.
Methods:
The methods section outlines the study's design, data collection, and statistical analysis. However, there are several issues that need attention:
1. Lack of Methodological Detail: The methods section lacks sufficient detail regarding the data collection process, questionnaire design, and participant recruitment. Readers should have a clear understanding of how the sample was selected and how the questionnaire was administered.
2. Statistical Analysis Explanation: The description of the statistical analysis is quite technical and dense. It would be helpful to provide a simplified summary of the analysis steps without overwhelming the reader with statistical terminology.
3. Missing Ethical Considerations: The methods section should include information about ethical considerations, such as approval from an institutional review board or ethics committee, and how informed consent was obtained from participants.
Reviewer's Report for Manuscript: "Association of Computer Vision Syndrome with Depression/Anxiety among Lebanese Young Adults: The Mediating Effect of Stress"
Results and Discussion:
The results and discussion sections of the manuscript provide valuable insights into the associations between Computer Vision Syndrome (CVS), mental health outcomes, and stress. However, there are some points that need attention and improvement:
1. Overemphasis on Statistical Information: The results section is heavily focused on presenting statistical values, p-values, and effect sizes. While this information is important, it tends to overshadow the interpretation of the results. The discussion should prioritize providing context and meaningful insights into the findings rather than solely presenting numerical data.
2. Insufficient Interpretation of Findings: The discussion section lacks a thorough and comprehensive interpretation of the results. The authors should provide more in-depth explanations of the observed associations between CVS, anxiety, depression, and stress. How do these associations align with existing literature? Are there any potential mechanisms underlying these relationships that could be explored further?
3. Limited Exploration of Implications: While the study establishes associations between CVS, mental health, and stress, the discussion could benefit from a more extensive exploration of the implications of these findings. How might these associations impact healthcare practices, public awareness, or policy decisions? Are there any potential interventions or strategies that could be derived from the study?
4. The Role of Stress as a Mediator: The study highlights stress as a mediator in the associations between CVS and mental health outcomes. However, the discussion does not delve into why stress might mediate these relationships. The authors should explore potential explanations for this mediation, considering psychosocial, physiological, and behavioral factors that could contribute.
5. Absence of Comparison with Prior Studies: The discussion should include a more detailed comparison of previous research findings. Highlighting similarities and differences between this study's results and previous studies on CVS, mental health, and stress would provide a broader context for readers to understand the significance of the findings.
6. Addressing Limitations More Thoroughly: The limitations section is appropriately mentioned, but some limitations could be discussed in more detail. For instance, the potential impact of the snowball sampling technique on generalizability and the absence of clinician diagnosis for symptoms should be further addressed.
7. Recommendations for Future Research: While the conclusion mentions the need for further studies to confirm the results and explore more factors and mediators, providing more specific recommendations for future research directions could be beneficial. What specific areas or aspects should researchers focus on to build upon these findings?
nil
Author Response
Abstract:
The abstract provides a comprehensive overview of the study's objectives, methodology, and findings. However, there are a few areas that need improvement:
- Lack of Clarity in Research Gap: While the abstract highlights the significance of the study in exploring the association between Computer Vision Syndrome (CVS) and mental health issues, the abstract fails to articulate the research gap clearly. Given the existing literature and what specific knowledge gap it addresses, it should explicitly state why this study is important.
We agree with the reviewer comment and we added the main gap:
With this excess use of electronics, computer vision syndrome (CVS), a highly prevalent condition, is associated with various symptoms. Although, understanding the relation between CVS and mental health has been reported, it did not explore the impact of CVS on more than one psychological aspect.
- Absence of Hypotheses: The abstract does not mention any hypotheses or expected outcomes. Including these would give readers a better understanding of the study's direction and focus.
We added the hypothesis for the abstract:
We hypothesize that higher CVS symptoms would be associated with higher levels anxiety, and depression, mediated by higher stress.
- Incomplete Mention of Findings: While the abstract discusses the associations between CVS, depression, anxiety, and stress, it lacks specific information on the strength of these associations and the extent of the mediating effect of stress. Quantitative results or effect sizes would enhance the abstract's clarity and impact.
We added the strength of the associations and the extent of mediating effect:
Stress partially mediated the association between CVS and anxiety and between CVS and depression. More CVS was significantly associated with more stress (Beta = 3.05). Higher stress was significantly associated with more anxiety (Beta = 0.70) and depression (Beta = 0.71) respectively, whereas more CVS was significantly and directly associated with more anxiety (Beta = 3.14) and depression (Beta = 2.61) respectively.
Introduction:
The introduction provides background information about Computer Vision Syndrome (CVS) and its potential impact on mental health. However, some issues need to be addressed:
- Limited Literature Review: The introduction lacks a comprehensive review of existing literature on CVS, mental health, and stress. A more detailed overview of relevant studies would provide a stronger foundation for the study's significance and research questions.
We added the following section to refer to the importance of our study:
CVS, a recently highly prevalent condition, was recently assessed for associated risk factors, the long-term effects, and related problems. However, our study is unique and the first of its kind worldwide to evaluate an association between CVS and mental health, with stress being a mediator in those associations. Analyzing the effect of stress is extremely important since if addiction to the electronic devices develop, stress proved to be a powerful mediator [33]. We hypothesize that higher CVS symptoms would be associated with higher levels anxiety, and depression, mediated by higher stress.
- Research Objectives and Hypotheses: While the study's objectives are briefly mentioned, they could be more clearly stated. Additionally, hypotheses should be explicitly outlined to guide readers in understanding the study's purpose and direction.
We added our hypothesis clearly as requested by the reviewer:
We hypothesize that higher CVS symptoms would be associated with higher levels anxiety, and depression, mediated by higher stress. And our objective is directly stated at the end of the introduction as follows: Therefore, the objective of this study was to determine the association between CVS with depression and anxiety among a sample of Lebanese young adults, along with evaluating the mediating effect of stress in these associations.
- Link to the Research Gap: The introduction should emphasize the unique contribution of this study in addressing the research gap. How does this study advance our understanding beyond previous research? This connection needs to be explicitly established.
We agree with the reviewer comment and we added the following:
CVS, a recently highly prevalent condition, was recently assessed for associated risk factors, the long-term effects, and related problems. However, our study is unique and the first of its kind worldwide to evaluate an association between CVS and mental health, with stress being a mediator in those associations. Analyzing the effect of stress is extremely important since if addiction to the electronic devices develop, stress proved to be a powerful mediator [33]. We hypothesize that higher CVS symptoms would be associated with higher levels anxiety, and depression, mediated by higher stress. In the absence of similar studies, we recommend further studies to assess those associations more in depth. Furthermore, most of the studies done were conducted in the Western region and examined the global prevalence but there is lack of studies which are conducted in Middle East and assessed the influence of CVS on mental health.
Methods:
The methods section outlines the study's design, data collection, and statistical analysis. However, there are several issues that need attention:
- Lack of Methodological Detail: The methods section lacks sufficient detail regarding the data collection process, questionnaire design, and participant recruitment. Readers should have a clear understanding of how the sample was selected and how the questionnaire was administered.
The study design has been changed to the following: This cross-sectional study, enrolling 749 participants, was conducted from August 2020 to April 2021, addressing digital device users in all of Lebanon’s districts (Mount Lebanon, Beirut, North, South and Bekaa) with an online questionnaire. The snowball sampling technique was employed to select participants for our research. To facilitate data collection, we developed a digital version of the questionnaire using Google Forms software, opting for an online approach for ease and accessibility. Before participation, all participants were informed about the study's objectives and provided with clear instructions for completing the questionnaire through an online platform. We rigorously examined Internet Protocol (IP) addresses to ensure data integrity to prevent duplicate survey submissions and maintain data validity. Notably, no incentives or rewards were offered to participants, ensuring that responses were entirely voluntary. Thus, participants were asked to send the link to others using a digital device. The questionnaires were written in Arabic and the link was sent via WhatsApp and email to participants.
- Statistical Analysis Explanation: The description of the statistical analysis is quite technical and dense. It would be helpful to provide a simplified summary of the analysis steps without overwhelming the reader with statistical terminology.
We deleted the extra details in the statistical analysis to simplify it to the readers.
- Missing Ethical Considerations: The methods section should include information about ethical considerations, such as approval from an institutional review board or ethics committee, and how informed consent was obtained from participants.
We agree with the reviewer comment and we added the ethical consideration.
Ethical Consideration
The ethical approval for this research was given by the Ethics and Research Committee of the Psychiatric Hospital of the Cross (HPC-034-2020). All participants provided their informed consent by submitting an online form. All research steps were conducted in accordance with the relevant guidelines and regulations.
Reviewer's Report for Manuscript: "Association of Computer Vision Syndrome with Depression/Anxiety among Lebanese Young Adults: The Mediating Effect of Stress"
Results and Discussion:
The results and discussion sections of the manuscript provide valuable insights into the associations between Computer Vision Syndrome (CVS), mental health outcomes, and stress. However, there are some points that need attention and improvement:
- Overemphasis on Statistical Information: The results section is heavily focused on presenting statistical values, p-values, and effect sizes. While this information is important, it tends to overshadow the interpretation of the results. The discussion should prioritize providing context and meaningful insights into the findings rather than solely presenting numerical data.
We agree with the reviewer comment and we added this section to provide meaningful insights into the findings:
Until this moment, this is the first study conducted among the Lebanese population evaluating the association between computer vision syndrome and mental health. Our pursuit of this topic is backed up by the fact that recent studies have pointed to a significant rise in CVS around the world during the past few years from 64% to 90%; thus, sparking our interest in evaluating the potential association between CVS and mental health. This current study was conducted to determine the association between CVS with depression and anxiety among a sample of Lebanese young adults, along with evaluating the mediating effect of stress in these associations.
- Insufficient Interpretation of Findings: The discussion section lacks a thorough and comprehensive interpretation of the results. The authors should provide more in-depth explanations of the observed associations between CVS, anxiety, depression, and stress. How do these associations align with existing literature? Are there any potential mechanisms underlying these relationships that could be explored further?
We agree with the reviewer comment and we added:
A prior study found that the presence of CVS was connected to higher anxiety levels, which can be explained by the connection between anxiety and physical discomfort due to eye illnesses, headaches, neck and shoulder pain [41]. Further studies linked the association between CVS and anxiety due to the cognitive over load. Extended use of computers required extensive concentration and tremendous attention which can contribute to mental fatigue and trigger feelings of anxiety. In addition, heavy use of electronic devices disrupts the sleep pattern, increases irritability, and heightens anxiety. Also, anxiety can be attributed due to the lack of physical activity and relaxation caused by the prolonged digital device use.
- Limited Exploration of Implications: While the study establishes associations between CVS, mental health, and stress, the discussion could benefit from a more extensive exploration of the implications of these findings. How might these associations impact healthcare practices, public awareness, or policy decisions? Are there any potential interventions or strategies that could be derived from the study?
We added the following section to the study implications:
Our results show that CVS is associated with psychiatric disorders such as depression, anxiety, and insomnia. The significant association between CVS and mental health may have important clinical implications for adult’s well-being and development. These results indicate the importance of early screening and assessments to identify individuals at risk for both physical and mental disorders. Thus, early intervention can help halt development of mental disorders and provide the necessary care. Furthermore, raising the awareness about the potential detrimental effect of digital devices is vital to implement proactive strategies. In addition, collaboration is needed on all levels basically between healthcare professionals and patients to address the health status and provide holistic support.
- The Role of Stress as a Mediator: The study highlights stress as a mediator in the associations between CVS and mental health outcomes. However, the discussion does not delve into why stress might mediate these relationships. The authors should explore potential explanations for this mediation, considering psychosocial, physiological, and behavioral factors that could contribute.
We added clarification for the mediating effect of stress:
It has been reported that stress medicate the association between CVS and depression and anxiety through physiological, psychological, and behavioral responses. Stress is considered a well-known trigger for the release of stress hormones as cortisol which disrupts the normal physiological hypothalamus-pituitary axis function including sleep regulation. These abnormalities contribute to physical discomfort associated with CVS and unstable mood. Furthermore, stress exerts effect on the cellular level manifested by the release of vasoconstrictors that affect the blood pressure, heart rate, and glucose levels which all exert an effect on cognitive function potentially deteriorating depression and anxiety.
- Absence of Comparison with Prior Studies: The discussion should include a more detailed comparison of previous research findings. Highlighting similarities and differences between this study's results and previous studies on CVS, mental health, and stress would provide a broader context for readers to understand the significance of the findings.
We compared our results to previous studies and the reason behind the association of depression, stress, and anxiety with CVS.
- Addressing Limitations More Thoroughly: The limitations section is appropriately mentioned, but some limitations could be discussed in more detail. For instance, the potential impact of the snowball sampling technique on generalizability and the absence of clinician diagnosis for symptoms should be further addressed.
We agree with the reviewer comments and we added the following to the limitation section:
Furthermore, our study population was homogenous (comprising solely Lebanese individuals) and using the snowball sampling technique, therefore limiting the generalizability of the findings to larger populations or different demographic groups.
- Recommendations for Future Research: While the conclusion mentions the need for further studies to confirm the results and explore more factors and mediators, providing more specific recommendations for future research directions could be beneficial. What specific areas or aspects should researchers focus on to build upon these findings?
We added specifically other factors to be studies in the future and highlighted in the below sentence:
Further studies are warranted to confirm our results and assess other factors such as the total time spent using any kind of digital screens, they type of electronic devices, and the reason for utilizing the device which might be considered as a vital factor that mediate the association between CVS and mental health.
Thank you!
Round 2
Reviewer 2 Report
The authors have diligently and effectively addressed the queries that were raised, resulting in a noticeable enhancement in the overall quality of the manuscript. As a result, I am pleased to recommend that the manuscript be accepted in its current form.
nil